# Polarimetric Imaging for Robot Perception: A Review

**DOI:** 10.3390/s24144440

**Published:** 2024-07-09

**Authors:** Camille Taglione, Carlos Mateo, Christophe Stolz

**Affiliations:** 1Vibot, ImViA UR 7535, Université de Bourgogne, 12 Rue de la Fonderie, 71200 Le Creusot, France; christophe.stolz@u-bourgogne.fr; 2ICB UMR CNRS 6303, Université de Bourgogne, 9 Avenue Alain Savary, 21078 Dijon, France

**Keywords:** computer vision, polarimetric, physic-based imaging, robotic perception

## Abstract

In recent years, the integration of polarimetric imaging into robotic perception systems has increased significantly, driven by the accessibility of affordable polarimetric sensors. This technology complements traditional color imaging by capturing and analyzing the polarization characteristics of light. This additional information provides robots with valuable insights into object shape, material composition, and other properties, ultimately enabling more robust manipulation tasks. This review aims to provide a comprehensive analysis of the principles behind polarimetric imaging and its diverse applications within the field of robotic perception. By exploiting the polarization state of light, polarimetric imaging offers promising solutions to three key challenges in robot vision: Surface segmentation; depth estimation through polarization patterns; and 3D reconstruction using polarimetric data. This review emphasizes the practical value of polarimetric imaging in robotics by demonstrating its effectiveness in addressing real-world challenges. We then explore potential applications of this technology not only within the core robotics field but also in related areas. Through a comparative analysis, our goal is to elucidate the strengths and limitations of polarimetric imaging techniques. This analysis will contribute to a deeper understanding of its broad applicability across various domains within and beyond robotics.

## 1. Introduction

The polarization of light, while not immediately apparent, is an important natural phenomenon. It manifests in different scenarios, such as the use of polarized sunglasses, LCD screens, medical imaging, and stress detection in materials. Analyzing this phenomenon often involves using Fresnel’s theory to understand how natural light reflects off surfaces [1,2,3]. Building upon this theory, numerous approaches have been developed, including the Jones calculus, Stokes parameters [4], Mueller calculus [5], Fourier analysis [6], least squares fitting [7], and more recently, Monte Carlo simulation [8] and machine learning [9]. These methodologies aim to extract diverse information from polarized light, facilitating its application across various domains. In robotic perception, polarimetric imaging is becoming increasingly important and continues to evolve. By leveraging the complex polarization state of light, polarimetric imaging enhances robots’ perceptual abilities, allowing them to detect subtle details that may be missed by traditional imaging methods. Polarization is gaining popularity in robotic perception, thanks to the proliferation of division-of-focal-plane (DoFP) sensors and advancements in deep learning techniques [10,11,12]. Unlike traditional cameras, polarimetric imaging offers a more profound understanding of the environment by analyzing the polarization properties of light. This reveals detailed information about surfaces, materials, and object features, enhancing robotic perception capabilities. Consequently, the polarimetric imaging community has explored the relationship between surface normals and light reflection to estimate shape distributions. Moreover, research in surface feature extraction [13,14,15] has further enriched the understanding of polarimetric imaging. A key focus of this community lies in 3D reconstruction, leveraging polarimetry to estimate 3D shapes by decoding polarization-encoded information in light [13,16,17,18,19,20,21]. By revealing subtleties such as surface texture, material composition, and object contours, polarimetric imaging empowers robots to attain a comprehensive understanding of their surroundings.

Polarimetric imaging holds potential for material discrimination. Research in estimating illumination and indices of refraction for different materials has gained interest [7]. Robots leverage the unique optical characteristics of materials to navigate environments more precisely. Furthermore, polarimetric imaging acts as a shield against visual distortion caused by reflections and glare. This technology helps robots equipped with polarimetric sensors to maintain clear visibility, even in difficult lighting situations. As a result, they can make better-informed decisions and perform tasks more efficiently. There exists a concerted effort within the polarimetric imagining community to refine image conditions, exemplified by endeavors to enhance contrast in images [22] for improved object visibility and ease of recognition. Some methods [23] focus on the suppression of reflections in images with high-specularity scenes. Furthermore, the utility of polarimetric imaging extends to image segmentation [24,25,26,27], an important aspect within the field of robotics, where precise delineation of objects is essential for tasks such as autonomous navigation and object manipulation. The efficacy of robots in object recognition and tracking experiences a substantial augmentation through the application of polarimetric imaging. By discerning the unique polarization characteristics associated with objects, robots can discern nuanced differences between similar surfaces, thereby refining the precision of their detection and tracking algorithms. Notably, polarimetric imaging, with its ability to capture and dissect the polarization state of light interacting with transparent or semi-transparent objects [13,24,28], emerges as a tool for obtaining detailed insights into their internal structure and composition. Unlike conventional imaging methodologies, which may encounter challenges in this regard, polarimetric imaging adeptly exploits phenomena such as birefringence and dichroism.

Incorporating polarimetric imaging directly into robotic systems promises to enhance the accuracy and reliability of current technological methods. This advancement drives robots toward higher levels of autonomy and adaptability across many applications. In the study of linear deformable object (LDO) perception, Zhaole et al. [29], integrating polarimetry, improve 3D reconstruction beyond RGB-D images, enhancing reliability and detail for better manipulation by robotic systems. Mora et al. [30] introduced a laser scanning method for identifying reflective surfaces. Integrating polarimetric imaging could improve its accuracy, helping in reflection data analysis, indoor environment modeling, and optimizing robot movements for safety and efficiency. Huy et al. [31] highlighted the importance of imaging and acoustics in underwater perception. Integrating polarimetry could enhance results by improving the signal-to-noise ratio (SNR) and object identification in low-visibility conditions. Wang et al. [32] could improve safety monitoring in low-light conditions by integrating polarimetric imaging. Even when incorporating polarimetric imaging has a lot of potential, with today’s technology we find challenging situations. For example, applications like [33] suffer from difficulties because of the low level of reflectance present on surfaces typically present in construction. But this application can still benefit polarimetry through 3D reconstruction of the walls to help detect cracks.

Polarimetric imaging finds itself important for applications across diverse domains such as remote sensing and underwater exploration. By dealing with surface reflections, polarimetric imaging develops the horizons of underwater navigation and marine ecosystem exploration, facilitating tasks ranging from the detection of submerged objects to ecological monitoring. Moreover, the impact of polarimetric imaging extends to the domain of bio-medicine, playing an important role in dealing with the intricate nature of biological complexity. By increasing contrast in tissues and structures, polarimetric techniques facilitate precise visualization in medical imaging applications. These advancements are driving progress in tissue analysis and diagnostic imaging. Instances of image segmentation utilizing polarimetry are common in medical imaging [5,34,35], facilitating the identification and classification of various tissue conditions, including healthy and cancerous tissues [35,36]. Bees orient themselves with sunlight polarization, and fish detect prey or avoid predators in low-visibility environments. Robotics research aims to replicate these abilities, with robots using polarization sensors for navigation [37,38]. In aviation, polarimetry could enhance guidance systems, especially in low-visibility conditions [39]. In addition, polarimetric imaging is a powerful tool for extracting structural and content information from biological tissues, complementing the extraction of material properties [35,40,41]. A more extended review of different state-of-the-art applications is shown in Table 1.

In the field of robotic perception, polarimetric imaging is currently rated at level 6 on the TRL (Technology Readiness Level) scale, or level 5 depending on the application. This means that the technology has been demonstrated in relevant environments, close to real operating conditions, or is at the prototype stage but has not yet reached sufficient maturity for widespread commercial use. At this stage, the community continues to refine the technology, resolve technical challenges, and validate its performance through practical field trials.

The rest of the paper is structured as follows. First, we discuss the fundamentals of polarization in Section 2. Then, in Section 3, Section 4 and Section 5, we review the state of the art on three concrete robotic perception problems improved with polarimetry: shape characterization for object recognition and manipulation; semantic segmentation for scene understanding and environment mapping; and pose estimation for object localization and manipulation planning in robotic applications. In addition, Section 6 summarizes the state-of-the-art applications in the robotics field using polarimetry. Next, we examine the performance of different state-of-the-art methods on segmentation and shape characterization in Section 7, and summarize current challenges in Section 8. Finally, we discuss the current solutions (Section 9) and conclude (Section 10) this review.

## 2. Principles of Polarization

Polarization imaging offers a rich source of information for computer visual applications, enabling advancements in scene understanding, material discrimination, and imaging under challenging conditions. Integrating polarization data with traditional imaging modalities leads to more robust and accurate computer vision systems across various domains. Wave light, as an electromagnetic wave, can be represented using the two perpendicular fields, electric and magnetic. Polarimetry represents the study of the oscillation of the orientation of the electromagnetic fields. Thus, exploiting the wave light properties we can obtain information like material properties or shape distribution. By definition, polarization describes the path traced by a light beam (random, linear, circular, or elliptical) in analogy with the modification of orientation and intensity of the electric field through time. Using the Fresnel equation, we can link material properties to the light that reflects from it [2].

The Fresnel theory is the first approach proposed to manage polarized light, an approach based on the Malus Law from 1808. This theory allowed us to understand for the first time the polarization phenomenon by studying the difference between the light before and after reflection on a surface. Reflections can be diffused in all directions, diffuse, or in a single direction, specular. Both components have different polarimetric properties. But concretely, the specular component is, in general terms, more polarized than the diffuse one.

### 2.1. Stokes Parameters

Stokes vectors are the most common approach to represent the state parameters of light polarization. Unlike the Jones vector, this mathematical representation can describe both partial and completely polarized light. Physically [60], a specific light beam can be described using the four-parameter vector $S={\left[S_{0}\;S_{1}\;S_{2}\;S_{3}\right]}^{⊺}$. According to Stokes in the work [4], a light beam can be modelled by the expression
$$I\left(\alpha ,\phi \right)=\frac{1}{2}\left(S_{0}+S_{1}\cos2\alpha +S_{2}\sin2\alpha \cos\phi -S_{3}\sin2\alpha \sin\phi \right),$$
which is written in terms of the four-parameter vector *S*. This equation describes the intensity of light observed through a linear polarizer rotated by an angle α with respect to a reference axis and a retarder with a phase shift ϕ. Thus, according to this model, the total intensity of light is encoded in the parameter $S_{0}$; the quantity of light polarized either horizontal or vertical is encoded in $S_{1}$; $S_{2}$ describes the light that is linearly polarized at $45^{\circ }$ or $135^{\circ }$; and $S_{3}$ describes the light that is circularly polarized. As Garcia et al., in [61], concluded, if the retardance ϕ is set to either $0^{\circ }$ or an angle of $90^{\circ }$ and the angle of linear polarization α is set to $\left[0^{\circ }\;45^{\circ }\;90^{\circ }\;135^{\circ }\right]$, the Stokes vector can be uniquely computed as
(1)$$S=\left\{\begin{array}{c} S_{0}=\frac{1}{2}\left(I\left(0^{\circ },\phi \right)+I\left(45^{\circ },\phi \right)+I\left(90^{\circ },\phi \right)+I\left(135^{\circ },\phi \right)\right),\;\phi \in \left[0^{\circ },180^{\circ }\right]\\ S_{1}=I\left(0^{\circ },0^{\circ }\right)-I\left(90^{\circ },0^{\circ }\right)\\ S_{2}=I\left(45^{\circ },0^{\circ }\right)-I\left(135^{\circ },0^{\circ }\right)\\ S_{3}=I\left(135^{\circ },90^{\circ }\right)-I\left(45^{\circ },90^{\circ }\right) \end{array}\right.,$$
where $S_{0}$ and $S_{1}$ represent the total and the difference of light intensity, respectively, between both components. $S_{2}$ and $S_{3}$ relate the phase difference between the horizontal and vertical components, providing information about the linear and elliptical polarization of the light.

### 2.2. Light Characterization Using Stokes Parameters

The three quantities determined from the Stokes parameter representation are the intensity, the degree of polarization (DoP), and the angle of polarization (AoP). While the intensity is codified directly by the first parameter $S_{0}$, DoP and AoP are computed as follows:(2)$$\left\{\begin{array}{c} \mathrm{DoP}=\frac{\sqrt{S_{1}^{2}+S_{2}^{2}+S_{3}^{2}}}{S_{0}}\\ \mathrm{AoP}=\frac{1}{2}\arctan\frac{S_{2}}{S_{1}} \end{array}\right.$$

DoP represents the proportion of the light that is polarized, where 0 indicates unpolarized and 1 represents linearly polarized. The degree of polarization can be used to obtain information about the material from which the light is reflected. In fact, if we create a map of the image size that for each pixel gives the degrees of polarization, then we can study the deformation of surfaces in the captured scene. AoP is the projection of the angle between the direction of the electric field and the angle of the polarizer onto the image plane (cf. Section 2.3).

### 2.3. Polarimetric Image Acquisition

The architecture of a polarimetric sensor consists of two key components: a light source emitting polarized light; and a polarization system with a linear polarizer adjusting light polarization. The interaction alters light polarization based on surface properties. Polarimetric sensors capture less light intensity compared to classical cameras due to the filtering process during acquisition. However, they provide complementary information on material properties. To leverage the strengths of different modalities, various approaches integrate polarimetric imaging with color imaging [26,51,62,63], shading [64,65], multispectral imaging [66,67], and depth sensing for object shape refinement [68,69]. Rotating a polarizer in front of a color camera was previously used to capture polarization information [70] but faced challenges such as time delays and complex acquisition setups, limiting real-time applications.

In recent years, polarization sensors gained popularity due to advancements in focal plane division sensors (DoFP) [10,11,12]. These sensors, comprised of micro-polarizers arranged in a grid, enable capturing multiple images of a scene with different polarizer angles in a single shot, without the need for internal mechanical rotation. This design allows for compactness and versatility in capturing images across various polarizations. The sensor’s architecture typically includes four Bayer filters corresponding to different polarizer angles, facilitating image acquisition. Depending on the combination of Bayer filters, the camera can be classified as standard color, gray polarimetric, or color and polarimetric. However, utilizing these sensors necessitates a pre-treatment phase known as demosaicking. The process described involves separating information from raw images using twelve channels, each representing polarizer angles and colors. Demosaicking is then applied to interpolate sparse images, filling gaps and ensuring uniform size. Various techniques like linear interpolation and machine learning, including deep learning and GANs, are utilized for this interpolation. Additionally, some methods leverage polarization properties or utilize adaptive filters for more accurate results. This improves traditional sensors because it captures scene images with a wider range of polarization angles, including commonly used angles like {0°, 45°, 90°, 135°}, as well as less common ones such as {0°, 30°, 60°, 90°} [15]. These diverse images enable the extraction of Stokes parameters (Section 2.2).

### 2.4. Weaknesses and Drawbacks

Polarimetry, like any measurement technique, faces challenges and uncertainties. These are caused by instrumental effects, noise, incomplete polarization description, and external factors like ambient light or interference [71]. Imperfect components and misalignment can contribute to ambiguity, as can failures during depolarization. Additionally, complex materials or samples with multiple scattering events or anisotropic properties can further complicate measurements. Understanding the sample’s nature is crucial for accurate interpretation in polarimetry.

The effect called “pi ambiguity” is relevant, relating to the cyclic nature of polarization measurements, which exhibit periodic behavior. This ambiguity effect poses problems for accurately determining the true polarization state of light, as several solutions may exist for a given set of measurements [72]. For example, by understanding this nature the calculus of surfaces via normal vectors can be obtained through the Snell–Descartes laws [73] of reflection and refraction. These laws link the angle of incidence $\theta_{i}$, the angle of reflection of light on a surface $\theta_{rc}$, the angle of refraction through the surface $\theta_{rf}$, and the normal to the surface (as reference frame).
(3)$$\begin{array}{c} \begin{array}{c} |\theta_{i}\left|=\right|\theta_{rc}|\\ n_{i}\sin\theta_{i}=n_{rf}\sin\theta_{rf} \end{array} \end{array}$$
where $n_{i}$ is the index of refraction of the first medium and $n_{rf}$ is the index of refraction of the second medium. But the method [15] used to obtain the normal vector depends on the calculation of zenithal and azimuthal angles, which involve a non-bijective function. This results in an ambiguous solution which provides two possible values for the azimuth angle. To resolve the ambiguity about the azimuth angle, several methods have been developed. Atkinson et al. [74] used shadow as an additional modality and diffuse reflections, Morel et al. [28] studied metal surfaces with active lighting, they constrained the zenith angle to solve the ambiguity in the azimuth angle. Stolz et al. [75] used multispectral imaging with the degree of polarization to resolve the ambiguity in the azimuth. Zhao et al. [76] developed a quadruple-polarized hybrid SAR system to resolve the azimuth ambiguity. Garcia et al. [61] studied the circular component of polarization to disambiguate the zenith angle, while Hwang et al. [16] used ellipsometry to resolve the azimuth ambiguity.

When calculating polarization angles and degrees using the Stokes vector, another ambiguity comes into play. The diffuse/specular ambiguity describes the direction of polarization that aligns parallel or perpendicular to the plane of incidence, depending on the dominant reflection (diffuse or specular). In diffuse cases, the degree of polarization (DoP) increases with the elevation angle, and the angle of polarization (AoP) aligns with the azimuth angle. In specular cases, the DoP increases up to the Brewster angle, after which it decreases. Diffuse reflectance involves scattering inside and transmission outside the surface, while the specular component is a direct reflection from the surface’s micro-facets. These components have a perpendicular angle of attack, and specular areas generally have a higher DoP than diffuse areas.

## 3. Image Segmentation Using Polarimetry

Segmentation is an important task of robot perception, particularly in domains like autonomous driving and robotic manipulation, where understanding the environment is also important for safe and effective operation. In autonomous driving, segmentation is integral to the perception system, enabling vehicles to interpret their surroundings in real time. Accurate identification of obstacles allows autonomous vehicles to navigate complex urban environments autonomously. Similarly, in robotic manipulation, segmentation is basic for robots to perceive and interact with objects efficiently. Whether in industrial automation or household robotics, robots must recognize and manipulate objects precisely. Segmentation aids in isolating objects from their backgrounds, facilitating tasks such as grasping, manipulation, sorting, and packing.

Often, segmentation suffers when dealing with transparent objects, reflective surfaces (water, glass, metal), and varying lighting conditions. Polarimetric imaging offers a promising solution by capturing both the intensity and polarization properties of light, providing valuable insights into object properties. Exploiting the polarization properties of light to extract detailed scene information, polarimetric imaging captures multiple polarization states, providing clues to deal with the segmentation of this kind of complex situation.

Traditional methods, such as thresholding, edge detection, region growth, active contours, Markov random fields, blob detection, region-based segmentation and machine learning with hand-crafted features, used hand-crafted rules and algorithms to identify and separate objects or regions of interest in images. Although these techniques are still used in scenarios with limited data or specific constraints, deep learning has become the predominant approach for image segmentation. Deep learning methods, with their ability to automatically learn features from data, have generally outperformed traditional techniques, particularly on large, complex datasets.

Most studies on segmentation tasks utilize the U-Net architecture [77], initially popularized in medical image segmentation. It comprises an encoder and decoder, enabling accurate segmentation. The encoder reduces spatial dimensions via convolutional and pooling layers, while the decoder increases them through oversampling and concatenation, refining the results. Notably, the architecture incorporates skip connections, linking the contracting and expansive paths, facilitating precise localization and enhancing model accuracy.

The U-Net architecture [25,27], renowned for its segmentation capabilities, is used in the processing of complex scenes with high accuracy and compatibility with data augmentation. Based on earlier models such as funnel convolution networks (FCNs) and fully convolutional networks (FCNs), U-Net overcomes the limitations of small-object processing while retaining spatial information, enabling multi-scale feature learning and facilitating real-time applications. Although other convolutional neural network (CNN) methods [24,49] have been used for segmentation tasks, U-Net remains the preferred choice due to its superior efficiency. Recent advances, such as attention mechanisms [24,51], have been incorporated into segmentation networks to improve performance by focusing on specific parts of the image. These mechanisms, borrowed from recurrent neural networks, transform neural networks to achieve better segmentation results [26].

Segment Anything [78] represents a big advancement in the field of image segmentation, allowing for precise identification of objects with minimal user intervention. A foundational deep learning model, SAM (Segment Anything Model), can interpret various types of prompts, including points, bounding boxes, and text descriptions, making segmentation accessible, intuitive and adaptable. This progress in segmentation technology holds promise for many applications, driving forward the capabilities of computer vision systems toward greater versatility and efficiency. This foundation can benefit from polarimetry, as can see in Figure 1, reducing segmentation errors by ignoring reflections and ensuring more precise results.

### 3.1. State-of-the-Art Methods in Polarimetric Segmentation

Here, we focus on the research in real-world application that utilize polarimetric imaging for segmentation in outdoor scenes for autonomous vehicles and segmentation of transparent objects for robotic manipulation tasks. Thus, we review works that exploit the potential of polarimetry to realize segmentation tasks.

Blanchon et al. [79] propose a new semantic segmentation approach based on a two-axis neural network architecture exploiting polarization cues. By exploiting these cues, the CNN is able to discern surface and material information, which are the two main objectives of their framework. The authors propose the regularization of surface normals as a means to delineate non-Lambertian regions, achieved via the incorporation of polarimetric features. To implement this concept, the authors utilize the projection of Stokes parameters onto the Poincaré sphere [80,81], enabling the extraction of key metrics such as intensity, AoP, and DoP. In addition, Blanchon et al. adopt Fresnel [3] equations to delineate the complex interaction between light and material properties, while modeling the polarization state of light in the scene based on the hue–saturation–luminance (HSL) framework.

Zhang et al. [26] proposed a cross-modal fusion method (Figure 2) for semantic segmentation, leveraging transformers. They employ parallel backbones for RGB and supplementary X-modal inputs, integrating cross-modal feature rectification modules and feature fusion modules. These rectify noisy data and uncertainties, utilizing attention-driven channel-wise feature modules and convolution for spatial weight maps. The method facilitates global information flow through a symmetric dual-path structure and cross-attention technique, ending in a final output feature map combining both modal routes. In both cases [26,79], the degree of linear polarization (DoLP) and the angle of linear polarization (AoLP) were used as representations of polarization data. The polarimetric datasets that were used by the authors were collected using a multi-modal vision sensor.

Kalra et al. [24] tackle the challenge of object instance segmentation for transparent objects, a task fraught with difficulties such as artefacts arising from anisotropic properties, light reflection, scattering, and the absence of texture. Leveraging polarimetric imaging, they introduce additional information to navigate these challenges effectively. Their methodology focuses on merging color and polarimetric data optimally to achieve accurate segmentation, utilizing the Mask R-CNN architecture widely employed for segmentation tasks. Their approach involves defining three input channels: gray scale images representing intensity; polarization angle; and degree of polarization. By integrating these modalities efficiently, Kalra et al. aim to enhance segmentation performance, particularly in challenging scenarios involving transparent objects.

While Yu et al. [82] push the boundaries of polarimetric image analysis by introducing a novel approach aimed at simplifying their architecture and eliminating the need for spatial attention patterns. Departing from recent methodologies [26,79], they adopt a polarimetric monochromatic imaging strategy, which allows for the separation of reflection and refraction within the image. Their primary contribution lies in the development of a transparent object segmentation network, which leverages extended interiors and enhanced edges while exploring various polarimetric representations and assessing their efficacy as network inputs. They notably emphasize the limitations of traditional representations like AoP and DoP, which are prone to noise and ambiguity; see Section 2.4.

### 3.2. Datasets for Polarimetric Image Segmentation

Different to more conventional modalities. Polarimetric-based robot perception has a lack of datasets in the literature. This is mainly due to two reasons, polarimetric imaging is a relatively recent modality in robotics, and the challenges that present polarimetric imaging in unstructured environments. But, we find already some initial approximations for the creation of polarimetric imaging datasets, like the two datasets presented in [25,51].

Refs. [25,27] introduced the Polabot dataset, tailored for polarimetric imaging of outdoor scenes, primarily intended for specular area segmentation. The dataset comprises 177 color images and their corresponding polarimetric equivalents, all captured using a multi-modal system featuring four cameras, including two color cameras with varying angles for stereo vision, a near-infrared camera, and a polarization camera. Despite the limited number of images, the authors proposed a data augmentation technique [83] to enhance the dataset’s utility for deep neural network applications.

Another popular dataset was developed by Xiang et al. [51]; it focuses on outdoor scenes for autonomous driving, utilizing an integrated multi modal sensor including polarization, RGB, infrared, and depth sensors. Their system captures color, infrared, polarization, and monocular depth images, with software providing direct output of stereo depth and surface normal information. In contrast to Polabot, their dataset offers higher resolution images (1024 × 1224) and consists of 394 images, expandable via augmentation techniques.

Creating a polarimetric dataset for autonomous driving involves several key steps. It begins with selecting suitable polarimetric cameras mounted on a vehicle to provide a 360-degree view. Planning diverse routes covering various driving scenarios and synchronizing cameras with LiDAR, radar, and RGB sensors are crucial. Post-collection data calibration and annotation are needed.

In this study, we opted to utilize the Polabot dataset, specifically focusing on the smaller-resolution variant. This decision was made to decrease the time and resources required for the evaluation process. Moreover, the Polabot dataset’s smaller resolution is highly compatible with the chosen augmentation method, which was developed by the same authors of the dataset. This compatibility ensures a better integration and potentially more accurate evaluation results, leveraging the augmentation techniques specifically tailored for this dataset.

## 4. Shape Characterization

Shape characterization is both a valuable tool and a challenge, encompassing tasks such as depth estimation, surface normal estimation, and 3D reconstruction. In robot perception, accurate shape characterization enables robots to comprehend and interact with their environment effectively, facilitating tasks such as navigation, object recognition, and manipulation.

### 4.1. Depth from Polarimetry

Traditionally, there has been many studies on recovering depth images from the geometry of either a stereo pair camera configuration or exploiting the geometry of structured light [84]. But these methods lack robustness most of the time when they are dealing with bad light conditions, textureless scenes, the presence of transparencies, reflections, etc. To address this problem, studies have recently focused on the potential of learning approaches [85]. Although they present potential benefits for dealing with situations like bad light conditioning, or textureless surfaces, they suffer from difficulties when dealing with other light interactions like reflectances or transparencies. Another way to improve on the limitations of traditional methods is to exploit other visual technologies, such as LiDAR [86]. But, although it is a technology that gives more accurate measurements, it is costly and weather-sensitive. In recent years, polarimetric imaging has emerged as a promising alternative for depth estimation in robotics [87] because the analysis of the polarization state of light reflected or scattered by objects in the scene helps deduce distances. By comparing these variations with known polarization patterns, accurate depth measurements can be obtained. This fact, is being also studied as a potential complementary modality [88], offering enhanced accuracy and robustness in certain scenarios.

A first approach to distance (depth) estimation using polarimetry was presented in [89], which focused on two parametric models involving electromagnetic plane waves. But these initial methods were impractical because of the need for controlling the scene and for prior knowledge of the material’s polarization characteristics. More recently, approaches like [68] utilize DoP and AoP, with calibration establishing a depth-to-polarization mapping. But still, linear polarized light sources tends to be necessary to calculate light depolarization post-interaction with the scene, as polarization changes with surface interactions.

To avoid the need of structured environments recent approaches propose the use of deep learning architectures for obtaining depth through polarization. These include recurrent neural networks (RNNs), which excel at processing sequential data but struggle with occlusion [90,91]. Fully convolutional networks (FCNs) process inputs of any size and produce outputs of the same size [92,93]. Encoder–decoder networks [94,95] are also employed for this task. Residual networks (ResNets) utilize residual connections for improved information flow between layers [96,97]. Siamese networks compare two inputs and generate a similarity score [98,99]. Additionally, radial basis function networks (RBFNs) [100,101] and attention modules have been explored for depth estimation through polarization [91,95].

### 4.2. Normal from Polarimetry

Normal estimation is a component of 3D shape characterization, fundamental in the field of robotics for tasks such as object manipulation, navigation, or scene understanding. Traditional methods for normal estimation, analogous to depth estimation, often rely on depth sensors or stereo vision, which may struggle in challenging lighting conditions or with transparent or specular surfaces. Normal-from-polarization is a method in computer vision and 3D imaging where normal information is extracted from polarized light [102] by analyzing measurements such as DoP and AoP, similar to depth estimation.

Today, deep learning schemes have emerged as the dominant approach. These include CNNs, which have demonstrated [103,104] robustness against variations such as rotation, occlusion, and different lighting conditions. However, CNNs may encounter limitations when dealing with time-series data.

On the other hand, radial basis function networks (RBFNs) have been employed to learn the non-linear relationship between polarization data and surface normals [105,106]. RBFNs utilize functions like Gaussian, multi-quadric, and inverse quadric as activation functions. While they offer fast training and perform well with non-linear problems, they are sensitive to input scaling, which can lead to issues if the input is not properly normalized.

Kirchengast et al. [107] proposed a LiDAR-based surface normal estimation method that can be enhanced by integrating polarimetric imagery. This improvement enhances point cloud accuracy, material differentiation, and overall LiDAR modeling robustness in diverse environments.

### 4.3. Three-Dimensional Reconstruction by Polarization

In the line of depth and normal estimation, works in the literature have addressed the problem of shape characterization is 3D reconstruction. This involves recovering the spatial structures of 3D objects from multiple 2D images or sensor measurements. Such 3D reconstruction is usually represented using voxel-based space [108,109], point cloud-based approaches [110], or multi-view based approaches [17,18,63]. But different to the depth and normal estimation, the problem of 3D reconstruction was rarely treated in previous works.

But with the advancements in polarimetric imaging, studies on 3D reconstruction have started to appear, making it a promising tool for various robotic applications, including autonomous navigation, object recognition, and scene understanding [14,15,21,28,63,75,111] because polarimetry offers a solution for capturing depth information in challenging lighting conditions like low light or glare [22,112,113]. Polarimetric imaging captures light polarization, revealing object orientation and surface characteristics, as discussed in the above subsections. Integrated with traditional methods, it enhances 3D model creation by providing more surface property information. This improves surface normal estimation, helping quality control and detailed 3D modeling, and enhances material classification. Polarimetry also offers robustness in challenging environments, vital for autonomous navigation in adverse weather. Integrating polarimetric imaging with traditional methods can enhance depth estimation accuracy and performance in textureless regions (transparent surfaces). Combining LiDAR with polarimetric data can create more detailed and reliable environmental maps.

Recent advancements in 3D reconstruction leverage the power of neural networks to exploit information encoded in polarized light. This field investigates the application of CNNs [114,115] for this purpose. While CNNs excel in non-linear mapping, their generalization capabilities can be limited [114]. As an alternative, generative adversarial networks (GANs) have demonstrated exceptional performance in generating 3D models directly from polarization images [19,116]. Notably, GANs exhibit robustness to noise and incomplete data [19,116] but face difficulties in making the method converge. For scenarios involving dynamic scenes, recurrent neural networks (RNNs) offer a compelling solution due to their effectiveness in temporal modeling [117,118]. Finally, a less standard architecture, deep belief networks (DBNs), holds promise for 3D reconstruction tasks, particularly those utilizing unlabeled data, by virtue of their ability to learn hierarchical representations from the input [119].

### 4.4. State-of-the-Art Methods for Shape Characterization

The discussed methods for normal estimation vary, with some employing non-learning techniques like non-linear algorithms [120,121], while others opt for learning-based approaches, utilizing various architectures [14,63,122].

Lei et al. [122] propose a method that enables scene-level normal estimation (Figure 3) from a single polarization image. Their approach combines data-driven learning with physics-based priors, allowing for robust normal estimation across various lighting conditions and scene types, both indoors and outdoors (Figure 4). By incorporating a multi-head self-attention module and viewing encoding (close to Nerf [123,124,125]), the method addresses challenges such as handling errors caused by scene complexity. They leverage intrinsic parameters of polarization cameras to determine viewing directions, which aids in resolving local ambiguities in polarization cues. Their network architecture, based on U-Net, processes raw polarization images to extract intensity, DOP, and AOP information.

Dave et al. [63] introduced Polarization-Aided Neural Decomposition Of Radiance Architecture (PANDORA), a method for reconstructing (Figure 5) object geometry and appearance from multiple images. This polarimetric inverse rendering approach utilizes implicit neural representations, offering a compact representation of signals. Polarimetric inverse rendering is a computational technique used to recover the physical properties of a scene, such as surface material properties and lighting conditions, from polarimetric observations collected by imaging sensors. Object shape and reflectance characteristics are modeled using coordinate-based neural networks, while incident lighting is represented through an implicit network with directional embedding. An important disadvantage of this method is that it requires specific conditions during acquisition, such as non-polarized incident lighting and opaque objects composed of dielectric materials.

Deschrainte et al. [14] developed a method (Figure 6) which estimates the 3D shape of objects along with spatially varying reflectance properties such as diffuse and specular albedo maps and specular roughness maps. They utilized polarimetric information extracted from the scene using a standard method, Section 2.3, and acquired images under frontal flash illumination from a single view direction. Their method employed an improved U-Net architecture to extract information from polarization images and two explicit cues: a reflectance cue representing normalized diffuse color and a shape cue in the form of a normalized Stokes map. The Stokes map calculates normalized changes in reflectance when viewed through different polarization filter orientations. Their network architecture employs an encoder–decoder structure with three specialized branches focusing on different aspects of shape and appearance: diffuse and specular albedo, specular roughness, and surface normal and depth.

Yu et al. [121] propose a technique for extracting surface height from polarimetric data. Unlike traditional methods, their approach focuses on non-linear optimization through direct minimization of disparities between predicted and observed intensities across all pixels and polarizer angles. It accommodates various illumination, reflectance, and polarization models, including a variant that eliminates the need for illumination and albedo information by utilizing image ratios. Surface height serves as the unknown variable in a non-linear least squares optimization approach. By measuring and optimizing the difference between actual data and projections based on predicted surface height, the authors achieve direct surface height estimation. They utilize intensity, AoP, and DoP as representations of polarized light and employ vector ratios between observed intensities for direct surface height estimation. The authors utilize a probabilistic polarization model and employ the best linear unbiased estimator to minimize error between observed ratios and model predictions.

Smith et al. [120] proposed a method to estimate surface height from a single polarization image. Their approach involves solving linear equations to estimate height, addressing local ambiguity in surface normal azimuth angles. This method allows for the estimation of spatially variable albedo or illumination solely from the polarization image, reducing the need for known lighting and albedo information. By converting polarization and shading constraints into linear equations, they provide globally optimal height estimation. The authors investigated the use of ambiguous surface normals and unpolarized intensity data for calculating illumination and per-pixel albedo, leveraging standard representations of polarized light [26,50,79] such as intensity, AoP, and DoP. They utilized a sparse QR solver algorithm to determine the optimal height map.

### 4.5. Datasets for Polarimetric Shape Estimation

As mentioned above, datasets associated with shape characterization methods generally impose more strict acquisition requirements than those used for segmentation with polarimetric imaging. Control over various conditions, like incident lighting, background settings, and diverse viewing angles, becomes imperative in order to ensure the accuracy and reliability of the method using the data. This controlled environment allows researchers to manipulate key factors influencing the rendering process, facilitating a nuanced understanding of the interplay between incident light, object geometry, and material properties.

Boss et al. in their work [126] present a synthetic dataset made for capturing the shape of an object and its appearance (SVBRDF). They use, 1125 high-resolution SVBRDF maps. The authors randomly resize and take 768 × 768 crops of these material maps. They apply random overlays together with simple contrast, hue, and brightness changes. In the end, they obtain 11,250 material maps. They use domain-randomized object shapes to apply a random material on nine different shape primitives (spheres, cones, cylinders). They also apply six to seven random materials to the scene. For environment illumination, they collect 285 high-dynamic ranges (HDRs). To render images, they use the Mitsuba renderer to create two-shot flash and no-flash images to obtain the maximum information for the appearance estimation [127]. In total, the dataset contains 100 K generated scenes.

PolarNet is another synthetic dataset developed by Deschaintre et al., as detailed in their work [14]. This dataset has garnered significant attention in the domain of inverse rendering applications due to its comprehensive nature and high-quality data. It comprises a rich collection of 100,000 synthetic images, each captured from four distinct polarizer angles. The key highlight of the PolarNet dataset lies in its provision of ground truth data for the SVBRDF (Surface–Volume Bidirectional Reflectance Distribution Function) attributes, including normal, diffuse, roughness, and specular components, alongside the depth map ground truth. To ensure realistic conditions, a random texture and object are selected for each image. Additionally, PolarNet offers supplementary cues such as the normalized Stokes map and a color diffuse map, enhancing the richness and diversity of the dataset. Deschaintre et al. employed their proprietary model, previously published in [128], utilizing the full Fresnel equation for the creation of the materials [73].

Creating a polarimetric dataset for 3D reconstruction involves steps like selecting suitable imaging equipment, typically a polarimetric camera with polarizers. Scenes are prepared with consistent lighting, and images are taken from various angles with different polarizer orientations. The collected data are processed to extract polarization information, including degree and angle of polarization. Advanced algorithms combine this with traditional methods for enhanced reconstruction detail.

In the context of shape characterization, the PolarNet dataset emerges as an ideal choice for comparing various methods. This preference is attributed to the exhaustive information provided by the authors, particularly with regard to the availability of ground truth data, which significantly aids in comprehensive evaluations and analyses. As a result, PolarNet continues to serve as a benchmark in the field, facilitating the assessment and advancement of various techniques and algorithms.

## 5. Pose Estimation Using Polarimetry

Another task studied in the literature on robotic perception is 6D pose estimation. Which involves determining the position and orientation of objects relative to a reference frame. Pose estimation comprises several different types: model matching [129], learning-based approaches [130], pose refinement [131], and multi-view systems [132], each adapted to robotics applications. Traditionally, pose estimation [130,133] relies on visual cues extracted from color, texture, or geometric features (RGB-D), but these methods can produce errors with transparent or reflective surfaces, as shown in Figure 7. However, these methods can be limited in challenging environments with low lighting, occlusions, or textureless surfaces. Polarimetry offers advantages by capturing not only the intensity, but also the polarization properties of light reflected from objects. This additional information can enhance the discriminative power of pose estimation algorithms, particularly in scenarios where traditional visual cues may be insufficient. Polarimetric data provide additional cues about object materials, surface properties, and geometric features, which can enhance the precision of pose estimation. Some methods, like D.Gao [45,46], have studied the 6D object pose from a monocular point of view. This hybrid model exploits physical a priori such as angle of linear polarization (AoLP), degree of linear polarization (DoLP), and normal maps to learn object transformations between the object image and the camera image. Polarimetric cameras offer additional information, enhancing surface and reflection analysis. Its network architecture comprises two encoders processing polarimetric images and physical normals separately, followed by merging and decoding to generate object masks, normal maps, and dense correspondence maps (NOCS). In particular, the model enables accurate pose estimation of highly reflective and transparent objects by integrating polarization properties with conventional intensity-based methods.

By accurately estimating the 6D pose of objects, robotics systems can effectively interact with the physical world, enabling tasks such as object grasping, object tracking, and scene reconstruction with higher precision and efficiency.

## 6. Experimental Robotic Applications Using Polarimetry

Polarimetric segmentation is gaining popularity in robotic perception due to its capacity to provide nuanced insights surpassing those of conventional RGB imagery, offering enhanced scene understanding and object delineation across various applications. In material characterization, it aids non-destructive testing for quality control across plastics, ceramics, and composites [134]. Three-dimensional reconstruction by polarimetry is also valuable for non-destructive testing [135,136], defect identification [134,137], stress analysis [138,139], and quality control in various industries, including aerospace and civil engineering. By analyzing the polarization state of reflected light, these methods can determine surface characteristics such as roughness, texture, and anisotropy [140,141], aiding in material characterization. Surface normals derived from polarimetry can assist in classifying different materials [134,142], crucial for industries like manufacturing and quality control. Additionally, in tracking and object recognition, depth polarimetric sensing and surface normal estimation provide cues about surface depth and orientation, enhancing computer vision systems’ ability to identify and track objects in challenging conditions [22,112,113] or with transparent or specular objects within the scene [13,118,143].

Polarimetric-based methods improve performance in object detection under challenging lighting [22,112,113] and adverse weather conditions [52,56], as well as in medical imaging for tissue classification [144,145]. Integration of polarimetric data with other modalities, such as RGB or multispectral imagery [146,147], further improves the segmentation models’ effectiveness across various applications. Especially, polarization-based depth estimation is particularly beneficial for underwater imaging [148,149,150] due to its ability to withstand light scattering and absorption, unlike traditional methods. This resilience enables precise depth information acquisition, crucial for underwater tasks such as archaeology, marine biology, and robotics. But also, in medical imaging, polarization-based surface normals contribute to techniques like endoscopy [151,152] and microscopy [40,145,153], assisting in detecting tissue abnormalities and aiding in diagnosis and visualization of pathological conditions for dermatologists and ophthalmologists. This advancement in imaging technology allows for non-invasive visualization of tissue structures and cellular details, benefiting medical diagnostics and endoscopy by improving organ visualization and aiding in disease detection, especially cancer [36,151,154]. Finally, we highlight that polarimetric imaging is a valuable technique used for analyzing transparent objects [24,155,156] across various fields, from scientific research to industrial applications. By assessing the polarization state of light, it allows for detailed visualization of transparent materials, which is difficult with traditional imaging methods. Applications range from microscopy in biological research to quality assessment of pharmaceuticals and detection of defects in materials like glass or plastics. Its ability to reveal subtle surface characteristics and internal structures makes it suitable for diverse applications where transparency poses challenges.

## 7. Experimentation

This section evaluates the different methods previously outlined. We examine their usefulness in the tasks reviewed above. Concretely, we have chosen the methods CMX [26] and Vibotorch [79] on the Polabot database [25,27] for segmentation evaluation. While the methods proposed by Lei et al. [122], Smith et al. [120], Yu et al. [121], Dave et al. [63], and Deschaintre et al. [14] were used for the evaluation of shape characterization on the PolarNet from [14].

### 7.1. Performance Evaluation of Polarimetry Segmentation

The first experiment was conducted over the CMX method [26] using two different architectures, SwinTransformers (SwinT) and Segformer (Segf), for image segmentation tasks. In addition, to the Segf architecture, we evaluated three different input modalities (Figure 8) color RGB, AoLP, and DoLP. To understand the potentials and limitations of each method, we use the accuracy, recall, F1 score, precision and intersection over union (IoU) metrics [157]. Accuracy shows the proportion of correctly classified instances among all instances. Recall presents the ability of the model to capture all positive instances. Precision discriminates the proportion of true positive predictions among all positive predictions. F1 score balances precision and recall metrics using the harmonic mean. Finally, IoU measures the overlap between predicted and ground truth regions in tasks like object detection and segmentation.

The results presented in Table 2 indicate that the Segformer architecture outperforms SwinTransformers when it is solely used color images as input. However, significant improvements are observed when the model uses the Segformer architecture with either polarization angle or polarization degree images. The effectiveness of this approach can be attributed to the cross-modal feature rectification (CM-FR) and feature fusion (FF) modules, which facilitate feature rectification and fusion between parallel streams at each stage of feature extraction. The CM-FR module employs a dual-rectification approach, incorporating both channel-wise and spatial-wise feature rectification, while the FF module further enhances information interaction and combination through a two-stage process involving information exchange and fusion.

In contrast, the results presented in Table 2, Table 3, and Table 4, relative to the experiment using the Vibotorch method [79] show consistent improvements when the model is trained using data augmentation. Augmented data not only aids in mitigating potential overfitting but also enhances the model’s ability to generalize across diverse instances within the dataset. Consequently, the findings underscore the efficacy of data augmentation strategies in bolstering the robustness and overall performance of the Vibotorch method, as indicated by the superior results attained under augmented conditions.

We also studied the intersection over union on common segmentation classes such as glass, cars, buildings, sky, and roads; Table 4. Here, it is observed that the CMX method of [26] performs better than the Vibotorch method of [79].

A more exhaustive evaluation can be read in [26]. The study explores the impact of different data modalities on grating performance, including colorimetric data alone and combinations with polarimetric information. Comparing their CMX (Segformer) to SwiftNet [158] on colorimetric data, they find that integrating polarimetry enhances segmentation accuracy, particularly for outdoor scene analysis in autonomous vehicles.

### 7.2. Performance Evaluation on Shape Characterization

To assess and understand the advantages and limitations of each of the evaluated methods (Lei et al. [122], Smith et al. [120], Yu et al. [121], Dave et al. [63], and Deschaintre et al. [14]), we use the mean angular error (MAE), median angular error (MedianAE) and root mean squared error (RMSE) metrics.

With regard to the studied shape estimation option (normal vector estimation) (results in Table 5) each of the studied methods have their own particularity and are not all used in the same context. While the methods of Lei [122], Smith [120], and Yu [121] focus solely on estimating the surface normal in the scene, Dave’s PANDORA method [63], on the other hand, provides an approach that estimates the diffuse and spectral components for 3D rendering and the normal object surface too. And Deschaintre’s method [14] proposes the estimation of diffuse, specular, normal, depth, and roughness components of the object (Figure 9). Each of these methods has been designed for use with a specific database, such as Lei’s work, which is used with indoor images with no control over scene illumination; or Deschaintre’s work, which required complex control over both scenes background and illumination conditions. Similarly, certain works, such as Dave’s, require more varied views of the object. For our analysis, we studied these methods on a common dataset, the one proposed by Deschaintre, because it is the most complete of those proposed, as mentioned in Section 3.1. It provides polarimetric images and field truths for many objects with different textures. We can also note that Smith and Yu’s methods are not based on a deep learning approach.

Non-learning-based methods like those of Smith et al. [120,121] have good accuracy. Learning-based approaches, such as PANDORA [63] and Deep SVBRDF [14], offer advantages in terms of stability and adaptability. These methods offer the best compromise between accuracy, stability, and generalizability in normal vector estimation tasks.

## 8. Open Challenges

Traditional polarimetric cameras used a mechanically rotating polarizer in front of the sensor, causing images with different polarizer angles to be captured at different times. This made simultaneous image acquisition impossible. The introduction of a new sensor simplifies the acquisition setup by removing the need for an internal mechanical rotation component. Careful illumination control is crucial for data quality. However, achieving precise control over lighting conditions may require a more complex setup. This could involve building an opaque enclosure with a controllable, polarizable light source positioned and oriented accurately to capture detailed incident light data. Such a setup allows for optimal polarimetric information in captured images.

Polarimetric imaging offers rich insights into scene properties through analysis of reflected light, but challenges remain for practical implementation. Despite the progress made in sensors, issues persist with noise and artefact management. Pre-processing steps, like demosaicking, Section 2.3, are often necessary for refining raw polarimetric data before integration into robotic perception applications. Different methods in the field utilize polarizer angle images, which are typically generated early in the process after demosaicking.

As seen in the literature (works like [83]) one of the main problems that can arise when using polarimetry with neural networks is the lack of data. The absence of data cannot be solved by standard data augmentation methods. This is the reason certain research focused on creating specific polarimetric data augmentation methods. It should also be noted that for the segmentation methods studied [26,79], the data acquisition setup is based on many cameras, and its installation on a vehicle can be costly compared with shape estimation methods, some of which only require a single camera. The approach from Deschaintre [14] needs heavy control of the lighting conditions (flash illumination), and outdoor illumination remains an open challenge.

Each formalism of polarimetry (Section 2) presents unique advantages and is chosen depending on the particular needs of the research or application. But, real-time applications still often require reduced image resolution due to the challenge of processing complex polarimetric imaging data, which involves multiple images and channels. Unlike colorimetric imaging, which typically involves a single image with three channels, polarimetric imaging demands significant computing resources for effective processing.

Nevertheless, challenges arise in contexts like autonomous driving, where controlling scene lighting is impractical, leading to difficulties in maintaining consistent and controlled lighting conditions.

## 9. Discussion

To deal with the above weaknesses, today, it is proposed in the literature to integrate polarimetric imaging with various conventional modalities, including colorimetric [26], depth [159,160], multispectral [146,161], and thermal imaging [159,162]. Combining these modalities expands the range of surface characteristics and opportunities for material composition analysis in a scene. When coupled with depth modalities like LiDAR or structured-light imaging, polarimetric imaging enhances 3D reconstruction and scene comprehension, particularly in challenging environments. Furthermore, integrating polarimetry with thermal imaging offers insights into material properties, heat dissipation, and thermal signatures, which are valuable for remote sensing and environmental monitoring. The fusion of polarimetric imaging with spectral techniques enhances material classification and identification, benefiting applications such as remote sensing, vegetation analysis [140], and biomedical imaging. The future of polarimetric imaging in robotic perception shows promise for innovative developments.

Deep learning techniques tailored for polarimetric data processing are set to improve efficiency and robustness. Real-time integration of polarimetric imaging into autonomous vehicles and robotics could revolutionize navigation, mapping, and environmental perception for enhanced safety and decision-making. Advancements in polarimetric data demosaicking present exciting opportunities to improve image resolution and extract richer information from polarimetric imaging systems. As polarimetric sensors evolve, there is a growing need for robust demosaicking algorithms tailored to polarimetric data’s unique characteristics. Key research areas include noise reduction, interpolation accuracy, artefact suppression, and fusion. Exploring machine learning techniques like deep learning and neural networks holds promise for accurately reconstructing complete polarimetric images from incomplete data. Additionally, integrating adaptive filtering methods, optimized color calibration, and sophisticated optimization techniques can further enhance demosaicking algorithms for polarimetric data [163,164].

In the method presented by Yu et al. [121] the potential of the integration of a hybrid diffuse/specular polarization model is discussed. This model is designed to capture both diffuse and specular reflectance, allowing for the estimation of albedo and lighting parameters. Furthermore, the authors explore the feasibility of incorporating this approach into a CNN for depth estimation using polarization data.

The results presented in Section 7 show a general out-performance of CMX [26] against Vibotorch [79]. This is due to the incorporation of attention modules and a cross-modal strategy that improves the efficiency of the method. However, this performance improvement requires additional computing and energy resources. While these methods offer advantages in terms of accuracy, their effectiveness is also underlined by increased stability and robustness in varied data domains. However, this skill requires significant additional data volumes and computing resources, which represents a trade-off in terms of resource allocation compared to traditional linear and non-linear optimization techniques.

## 10. Conclusions

Polarimetric imaging addresses challenges like specular reflections encountered in robot perception, where traditional sensors struggle, by selectively capturing polarized light to enhance segmentation and depth estimation accuracy. It also improves transparent object detection by leveraging light polarization properties, aiding in perceiving and interacting with objects like glass panels or liquids. Additionally, in low-texture environments, polarimetric imaging provides meaningful depth and surface normal information, enhancing perception and navigation capabilities.

Integration of polarimetric imaging with conventional modalities enhances robotic perception by expanding surface characterization and material analysis within scenes. This fusion improves 3D reconstruction, scene comprehension, and provides insights into material properties, thermal signatures, and spectral characteristics. Advancements in polarimetric data processing, especially through deep learning, promise efficiency and robustness in applications like autonomous navigation and environmental monitoring. Recent advancements in shape characterization, particularly leveraging deep learning, offer improvements in accuracy and robustness, although they require substantial computational resources and datasets. These approaches extend beyond shape estimation, with potential applications in robotic manipulation and scene understanding, showcasing the versatility of deep learning-based methods.

The continuous evolution of polarimetric imaging and deep learning methodologies holds promise for advancing robotic perception. By addressing challenges like specular reflections, transparent object detection, and low-texture environments, polarimetric imaging enables more accurate perception and interaction with the surroundings. Integration of these technologies into robotic perception systems is expected to drive advancements in autonomous navigation, object manipulation, and scene understanding.

## Figures and Tables

**Figure 1 sensors-24-04440-f001:**
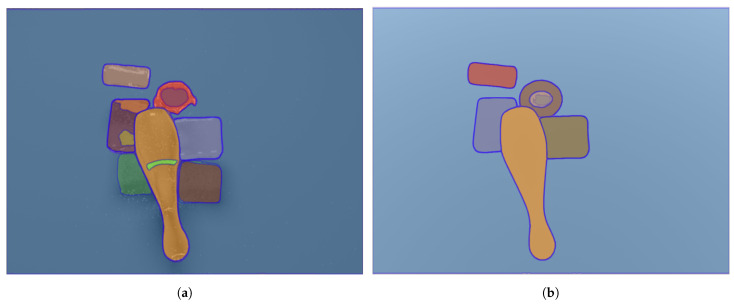
SAM results over-segment all reflection areas differently and also over-segment the shadow of the object. In this context, reflection removal using polarimetry is useful. (**a**) Segmentation results with only RGB (**b**) Segmentation results with both modalities.

**Figure 2 sensors-24-04440-f002:**
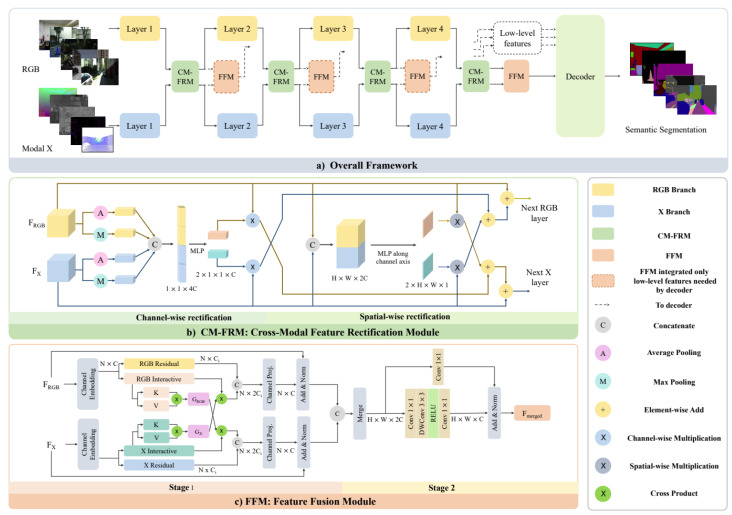
Architecture framework from CMX: Cross-Modal Fusion for RGB-X Semantic Segmentation with Transformers (source image [26]).

**Figure 3 sensors-24-04440-f003:**
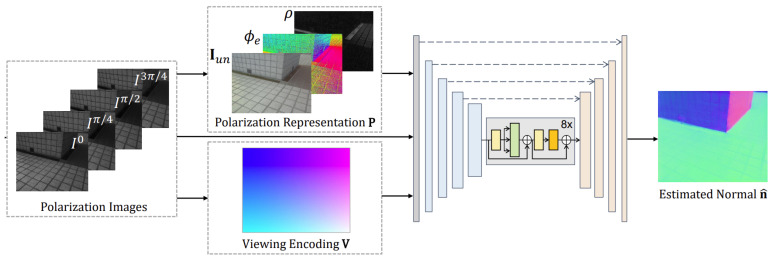
Architecture from Shape from Polarization for Complex Scenes in the Wild (source image [122]).

**Figure 4 sensors-24-04440-f004:**
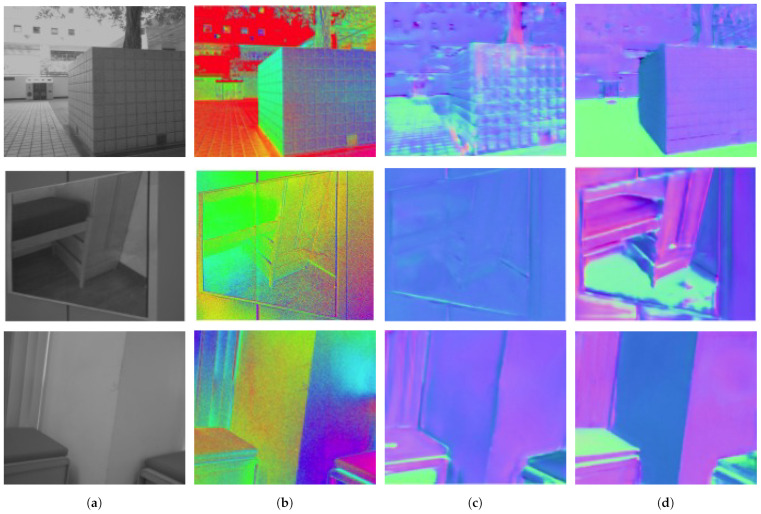
Normal estimation with and without polarization, (source image [122]). (**a**) RGB (**b**) AoLP (**c**) Normal from RGB (**d**) from RGB + AoLP.

**Figure 5 sensors-24-04440-f005:**
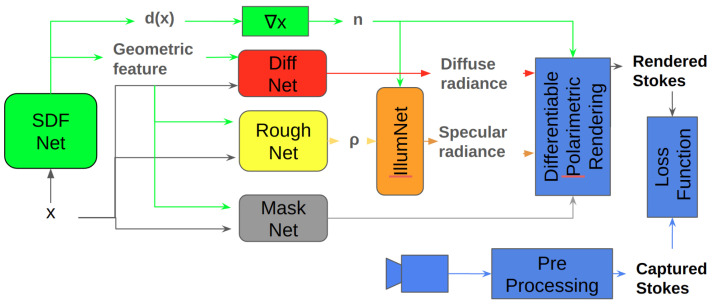
Polarization-Aided Neural Decomposition of Radiance Architecture (PANDORA) [63].

**Figure 6 sensors-24-04440-f006:**
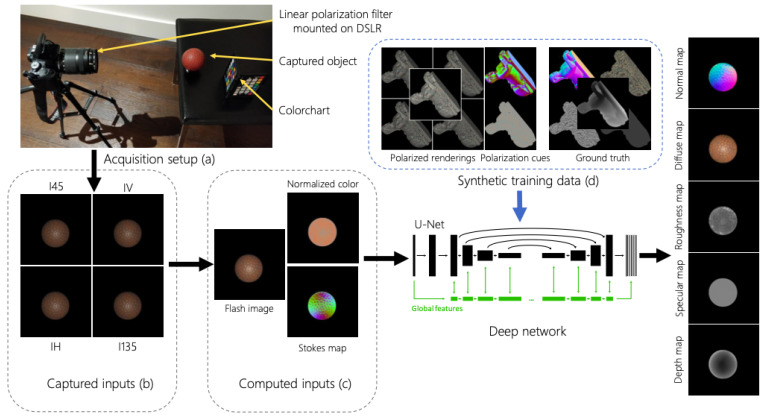
Deep Polarization Imaging for 3D shape and SVBRDF Acquisition global architecture (source image [14]).

**Figure 7 sensors-24-04440-f007:**
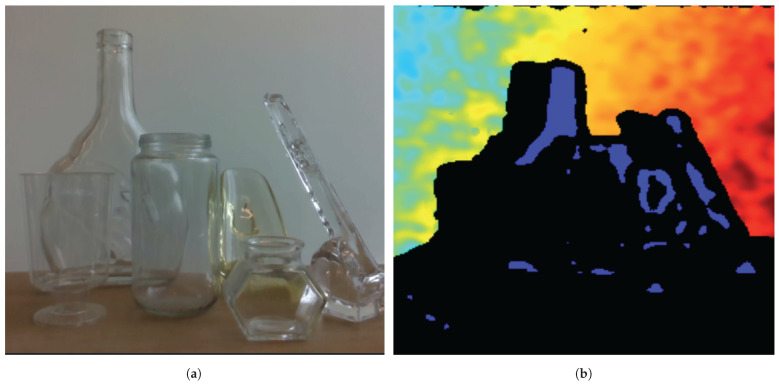
Depth image estimation from an RGBD sensor dealing with transparent objects. (**a**) Image acquired from the RGB lens (**b**) Image retrieved from the depth lens represent in jet colormap where blue color represents small distance and red ones larger distances.

**Figure 8 sensors-24-04440-f008:**
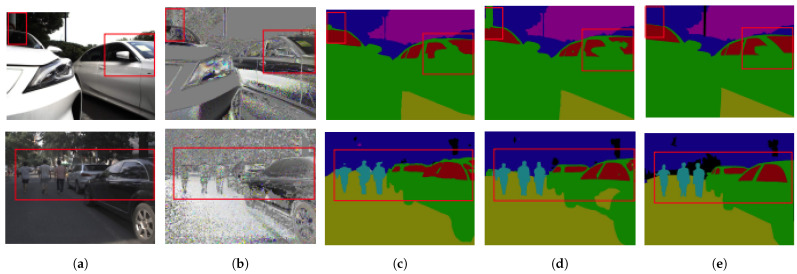
Two examples of segmentation for autonomous driving when using polarimetric imaging. Red boxes represent the areas of interest, where it is observed the improvement when dealing with reflective surfaces (source image [26]). (**a**) RGB (**b**) AoLP (**c**) Results from RGB (**d**) Results from RGB+AoLP (**e**) Ground truth.

**Figure 9 sensors-24-04440-f009:**
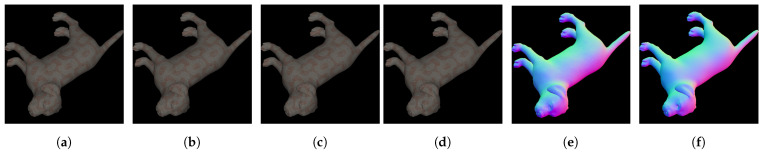
Normal estimation results (image source [14].) (**a**) Stock parameter I0 (**b**) Stock parameter I45 (**c**) Stock parameter I90 (**d**) Stock parameter I135 (**e**) Normal GT (**f**) Estimated normal.

**Table 1 sensors-24-04440-t001:** Table of different fields of application of polarimetric imaging.

Application	Condition	Papers
Transparent object	3D reconstruction	[13,42]
	Segmentation	[24,43]
Multi-fields	Shape estimation from diffuse polarization	[2]
	Surface and shape reconstruction by inverse rendering	[14,15,44]
	Reflection removal	[23]
	Reconstruction of metallic surfaces	[28]
	Pose Estimation	[45,46]
	Navigation	[37,38,39]
Medical	Mueller microscopy	[40,41]
	Robotic imaging	[47]
	Semantic segmentation	[48]
Autonomous driving	Semantic segmentation	[25,26,27,49,50,51]
	Object detection under bad weather	[52]
Landscape imaging	Haze removal	[51,53,54,55,56,57]
Industry	Piece inspection	[58]
	Sorting by semantic segmentation	[24]
Low-light applications	Image restoration	[22,59]

**Table 2 sensors-24-04440-t002:** Evaluation results of experiment CMX using 3 different modalities (Color, AoLP, and DoLP) over 2 different architectures (SwinT and Segf).

	CMX SwinT	CMX Segf		
Input	RGB	RGB	AoLP	DoLP
mIoU	0.496	0.543	0.732	0.757

**Table 3 sensors-24-04440-t003:** Evaluation results of experiment with Vibotorch using 4 different conditions.

	Pre-Train		Non Pre-Train	
	Non-Augmented	Augmented	Non-Augmented	Augmented
Accuracy	0.766	0.831	0.743	0.809
Recall	0.772	0.826	0.729	0.809
F1-Score	0.769	0.817	0.736	0.802
Precision	0.770	0.821	0.732	0.806
mIoU	0.3143	0.4426	0.2971	0.3857

**Table 4 sensors-24-04440-t004:** Evaluation results for the experiment CMX vs. Vibotorch using mIoU metric. Here, the performance of segmenting 5 different classes is evaluated.

	Conditions	Glass	Car	Sky	Road	Building
CMX Segf	RGB	0.57	0.71	0.68	0.75	0.73
CMX SwinT	RGB	0.54	0.66	0.63	0.74	0.60
CMX Segf	AoLP	0.73	0.78	0.76	0.81	0.76
CMX Segf	DoLP	0.74	0.78	0.75	0.80	0.76
Vibotorch	Non-Augmented/pre-train	0.11	0.42	0.28	0.32	0.41
	Augmented/pre-train	0.30	0.44	0.39	0.41	0.48
	Non-Augmented/Non-Pre-train	0.19	0.24	0.22	0.29	0.39
	Augmented/Non-Pre-train	0.16	0.31	0.21	0.26	0.34

**Table 5 sensors-24-04440-t005:** Evaluation results for the experiment on normal vector image estimation. Each metric (MAE, MedianAE, and RMSE) is with respect to the 3-component normal vector.

	MAE	MedianAE	RMSE
SFP-Wild [122]	47.3	32.47	120.4
PANDORA [63]	37.4	25.2	93.5
Deep SVBRDF [14]	36.6	9.43	119.8
SFP non-linear [121]	24.2	26.67	230.9
Height from polarization [120]	15.68	27.2	474

## Data Availability

The segmentation data used in this study are available in the public domain at the following resource: [https://search-data.ubfc.fr/FR-18008901306731-2022-05-13-02_PolaBot.html#data_deposit] accessed on 25 June 2024. The reconstruction data used in this study are available in the public domain at the following resource: [https://wp.doc.ic.ac.uk/rgi/project/deep-polarization-3d-imaging/] accessed on 25 June 2024.

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
