# Peer review of "Polarimetric Imaging for Robot Perception: A Review"

_sensors, 2024, doi:10.3390/s24144440_

Round 1

Reviewer 1 Report

Comments and Suggestions for Authors

This manuscript presents a review on Polarimetric Imaging for Robot Perception. However, there are several weaknesses in the current document that need to be addressed. The issues can be summarized as follows:

1. The author is advised to reduce the introduction of conceptual content and to concisely and effectively highlight the limitations of various studies throughout the paper, as well as emphasize the necessity of introducing polarization imaging technology. For example, before section 3.1, the method of segmentation research was introduced, but there is a lack of in-depth analysis of the limitations of related research, which is not enough to support the advantages or necessity of research based on polarization detection technology.

2. In Section 1, it is recommended to introduce more outdoor robotics-related papers for analysis to demonstrate the necessity of applying polarization technology, such as "3D vision technologies for a self-developed structural external crack damage recognition robot; Automation in Construction."

3. The author should introduce the performance gain results of the system after the introduction of polarization imaging technology. For example, in section 3.1, although the author stated the method of using polarimetry technology for segmentation, there was no description of the performance improvement of the model after the introduction of polarimetry technology, which cannot provide scholars with intuitive model optimization ideas. Similar problems exist in other sections.

4. In section 3.2.2, popular existing datasets are introduced, but the author did not explain the situation of the datasets used in existing research. Are existing studies all based on the datasets introduced, if not, it is recommended that the author introduce the key points or methods of dataset construction to provide new scholars in this field with a more comprehensive idea.

5. Some sections seem more like a brief introduction to the application scenarios of polarimetry technology, and do not reflect the progressive relationship between similar studies or the complementary situation in terms of technology, such as sections 4.1-4.3 and section 5.

6. The entire article lacks visual materials, especially section 7 should have sufficient qualitative and quantitative evaluations.

Comments on the Quality of English Language

After the major revisions, it is advisable to check the details of the manuscript.

Author Response

*** Comments and Suggestions for Authors ***

This manuscript presents a review on Polarimetric Imaging for Robot Perception. However, there are several weaknesses in the current document that need to be addressed. The issues can be summarized as follows:

  1. The author is advised to reduce the introduction of conceptual content and to concisely and effectively highlight the limitations of various studies throughout the paper, as well as emphasize the necessity of introducing polarization imaging technology. For example, before section 3.1, the method of segmentation research was introduced, but there is a lack of in-depth analysis of the limitations of related research, which is not enough to support the advantages or necessity of research based on polarization detection technology.

Authors response:

We appreciate this valuable comment. In order to clarify this point, we made two modifications in the original manuscript. Firstly (A) we have modified the introduction following the reviewer's advice, and secondly (B) clarify section 3 modifying the text and adding some supporting illustrations.

(A) To simplify the conceptual content of the original paper, we have removed the equations detailing the transformation of electronic wave polarization into polarimetric light, as well as the more theoretical aspects of the Fresnel equations. Instead, we refocused the introduction to highlight practical applications within the field of robotic perception (lines 66-78), emphasizing their relevance and impact.

(B) Furthermore, we have improved the segmentation, depth estimation, and 3D reconstruction sections adding references ([34-36], [52-56], [79], [85-91], [104], [109], [131-135]) that explore different techniques to solve different challenges in those domains without the incorporation of polarimetric imaging, illustrating the potential advantages of integrating this modality.

Specifically, we incorporate a new illustration to exemplify the contribution of polarimetric imaging. For example, we compare segmentation (Figures 1 and 3) and normal estimation (Figure 2) results, both with and without polarization.

Through the different illustrations added in the paper (Figure 1-3, 11), we showed how polarimetric imaging can mitigate the error in many contexts (reflective and transparent surfaces) for segmentation, 3D reconstruction or other tasks, where the color and depth imaging will bring error or lack of information.

  1. In Section 1, it is recommended to introduce more outdoor robotics-related papers for analysis to demonstrate the necessity of applying polarization technology, such as "3D vision technologies for a self-developed structural external crack damage recognition robot; Automation in Construction."

Authors response:

In the introduction (lines 67-77), we integrated references from articles outside the field of polarimetry that illustrate potential synergies with this modality.

For instance, "Intensity-Based Identification of Reflective Surfaces for Occupancy Grid Map Modification"([53]) demonstrates how the accuracy of laser scanning could be significantly enhanced through the incorporation of polarimetric imaging.

Similarly, in "A Robust Deformable Linear Object Perception Pipeline in 3D: From Segmentation to Reconstruction,"([52]) the authors highlight the potential for improving 3D reconstruction by incorporating polarimetric information.

Furthermore, in the context of underwater environments, "Object perception in underwater environments: a survey on sensors and sensing methodologies"([54]) identifies methodologies that could leverage polarimetric imaging to enhance sensing capabilities.

Lastly, "An adaptive image enhancement approach for safety monitoring robots under insufficient illumination conditions"([55]) suggests that polarimetric imaging could serve as a method to augment image contrast, thereby improving safety monitoring in challenging lighting conditions.

Concerning the work “3D vision technologies for a self-developed structural external crack damage recognition robot; Automation in Construction”([56]) today is a challenging domain of application for polarimetry because it should be covered the surface to study with a layer of reflective material. But we have included this reference to illustrate situations where polarimetric imaging has potentials but is still challenging. To clarify this, we have added the following text in the manuscript:

“Even when incorporating polarimetric imaging has a lot of potential, with today's technology we find challenging situations. For example, applications like~\ref{Hu2024} will suffer from difficulties because of the low level of reflectance present on surfaces typically present in construction. But this application can still benefit polarimetry through 3d reconstruction of the walls to help detect the crack.”

  1. The author should introduce the performance gain results of the system after the introduction of polarization imaging technology. For example, in section 3.1, although the author stated the method of using polarimetry technology for segmentation, there was no description of the performance improvement of the model after the introduction of polarimetry technology, which cannot provide scholars with intuitive model optimization ideas. Similar problems exist in other sections.

Authors response:

We agree with the reviewer comment that more exhaustive quantitative results will help to better illustrate the potentials of polarimetric imaging. Unfortunately, because of time constraints, we can not further extend the evaluation performed in section 7 (Experimentation) where different methods are evaluated.

Instead, to help the reader  better understand the potentials of polarimetric, we have added  new figures to the manuscript. We’ve included images illustrating the segmentation performance of the "CMX: Cross-Modal Fusion for RGB-X Semantic Segmentation with Transformers" method, both with and without polarimetry imaging, specifically highlighting its enhanced capability in accurately handling specular areas (Figure 1).

Furthermore, we incorporated additional images demonstrating the effectiveness of the "Shape From Polarization for Complex Scenes in the Wild" method for normal estimation (Figure 2). These images vividly depict how polarimetric imaging significantly enhances the accuracy and precision of the normal estimation process.

  1. In section 3.2.2, popular existing datasets are introduced, but the author did not explain the situation of the datasets used in existing research. Are existing studies all based on the datasets introduced, if not, it is recommended that the author introduce the key points or methods of dataset construction to provide new scholars in this field with a more comprehensive idea.

Authors response:

We thank the reviewer for this valuable remark. Effectively, the previous version of the manuscript was a bit ambiguous at the moment of showing the state of the art in dataset and benchmarking for polarimetric imaging. In fact, different from more conventional image processing, polarimetry is underrepresented, and specially in image segmentation. To make it more clear that we have rewrite the manuscript and we have added the text colored in blue at the beginning of section 3.2  (lines 332-336).

And following the suggestions of the reviewer we have introduced a description of today’s procedures to create datasets for polarimetry. More specifically, we have the attention on the segmentation domain and normal estimation

These updates provide an overview of today’s methodology presented in the literature for selecting and creating data sets.

  1. Some sections seem more like a brief introduction to the application scenarios of polarimetry technology, and do not reflect the progressive relationship between similar studies or the complementary situation in terms of technology, such as sections 4.1-4.3 and section 5.

Authors response:

We thank the advice of the reviewer, and we have restructured the three sections to keep the introductory essence of these subsections while we give the progressive relationship between similar studies and complementarity of technologies.

Now section 4.1 “Depth from polarimetry” after being completely rewrite, starts enumerating the different technologies used in the literature to produce depth (distance) measurements. After that, we put attention to the polarimetric technology and how it is being used until today, starting from signal analysis until data driven approaches.

Section 4.2 “Normal from polarimetry” we have shortened and rewrite the section to avoid repetitions with previous subsection and rewrite the current state of the art on methods about normal estimation.

We have to compact and rewrite Section 4.3 “3D Reconstruction by polarization” to give continuity with the previous subsections. 

Section 5 “Pose estimation using polarimetry” We have rewritten this section to arrange, and added further references ([131-135]) to better present the different types of pose estimation methods that use polarimetry or not.

In addition to the restructured text, we have added new references that support the progressive analysis of the evolution of each one of the problems. We would like to highlight that these revisions point to the emerging role of polarimetric imaging in advancing these fundamental computer vision tasks.

  1. The entire article lacks visual materials, especially section 7 should have sufficient qualitative and quantitative evaluations.

Authors response:

In order to address this point, we have incorporated image to support qualitatively the discussion in both segmentation tasks and normal estimation.

For each method ([14], [26], [60], [80]) under investigation, we have provided a comprehensive visual comparison.

These figures (Figure 1-2, 6-7) illustrate the used input images by each method, alongside the corresponding ground truth and the resulting output images generated by the methods.

Moreover, to highlight the advantages of polarimetric imaging, we have included specific images that demonstrate its effectiveness.

These images showcase method results or depth estimations on surfaces that are reflective or specular.

By presenting these qualitative comparisons (for example Figure 3), we aim to provide a clearer understanding of the strengths and limitations of each method, particularly in the context of reflective surface analysis.

Through this visual material, we believe that we address the remark of the reviewer, by  offering a more nuanced and comprehensive insight into the performance and applicability of the studied methods, thereby contributing to a deeper appreciation of the role of advanced imaging techniques in complex visual tasks.

*** Comments on the Quality of English Language ***

After the major revisions, it is advisable to check the details of the manuscript.

Reviewer 2 Report

Comments and Suggestions for Authors

Polarimetric imaging is a relatively new sensing technology that offers hope for handling difficult cases, such as shiny or transparent objects. Thus, any paper reviewing the current state of this technology is beneficial as it stimulates further development of the technology. The submitted manuscript serves this purpose well, especially since it tries to discuss the use cases in robotic applications.

The manuscript deserves to be published in Sensors after some issues listed below are addressed.

1) The authors provide an impressive list of up-to-date references. Many quoted papers refer to deep learning or machine learning (DL/ML) methods. However, in the segmentation section, there is a conspicuous lack of references to the foundation models, for example, Segment Anything Model (A. Kirillov et al., "Segment Anything," arXiv:2304.02643v1 [cs.CV]). This is not just another incremental improvement in the plethora of existing segmentation techniques but a significant game changer that should be quoted (even if there are no published examples of using foundation models to process polarimetric images).

2) The title of section 6 “Real-world robot application using polarimetry” is a bit misleading. There are, in fact, no “real-word” (aka commercial-ready) examples quoted in this section. While some references (e.g., [23],[106],107]) provide very interesting and encouraging experimental results, they are still at the research level. Please either tone down (change) the title or provide additional references supporting the title.

3) Related to item 2): while sections 2.4, 8-10 provide sobering comments about the current state of the polarimetric imaging, it could be helpful to include a general comment about the technology readiness level (TRL) in the introduction. From the manuscript, a reader may infer TRL = 5 or 6 at most (this covers hardware and relevant software).

Minor issues:

4) Please check if all referenced papers have a complete and correct quotation; some are incomplete (e.g., [25], line 799).

5) Please check the language and rephrase where needed, e.g. line 668: “This lack of data cannot be overcome by standard data augmentation methods, which is why some work aims to augment polarimetric data.”

6) It is rather unusual to report in the review paper the unpublished results of own ongoing research (not as a quotation of previous work but in the form of tables with actual results). The authors may consider whether extracting section 7 Experimentation and publishing it as a separate research paper (letter or full article) may be a better way to show their work.

Comments on the Quality of English Language

Please see item 5) in main Comments section.

Author Response

*** Comments and Suggestions for Authors ***

Polarimetric imaging is a relatively new sensing technology that offers hope for handling difficult cases, such as shiny or transparent objects. Thus, any paper reviewing the current state of this technology is beneficial as it stimulates further development of the technology. The submitted manuscript serves this purpose well, especially since it tries to discuss the use cases in robotic applications.

Authors response:

We acknowledge this reviewer's comment, this encourages us to keep working in this direction.

The manuscript deserves to be published in Sensors after some issues listed below are addressed.

1) The authors provide an impressive list of up-to-date references. Many quoted papers refer to deep learning or machine learning (DL/ML) methods. However, in the segmentation section, there is a conspicuous lack of references to the foundation models, for example, Segment Anything Model (A. Kirillov et al., "Segment Anything," arXiv:2304.02643v1 [cs.CV]). This is not just another incremental improvement in the plethora of existing segmentation techniques but a significant game changer that should be quoted (even if there are no published examples of using foundation models to process polarimetric images).

Authors response:

Initially, we didn’t consider semantic anything because of the lack of explicit relation with polarimetry. But as the reviewer suggested, this is a non-negligible step in the problem of segmentation. Because of this, we've considered adding a mention of "Segment Anything"[79] (lines 273-279) and highlighting how this advanced model revolutionizes segmentation with its high accuracy and versatility in various tasks.

Additionally, we discussed the benefits of polarimetric imaging, which reduces segmentation errors by ignoring reflections, ensuring more precise results.

Integrating "Segment Anything" with polarimetric imaging combines the strengths of both, enhancing image analysis accuracy and expanding applications in fields like autonomous driving, medical imaging, and remote sensing.

2) The title of section 6 “Real-world robot application using polarimetry” is a bit misleading. There are, in fact, no “real-word” (aka commercial-ready) examples quoted in this section. While some references (e.g., [23],[106],107]) provide very interesting and encouraging experimental results, they are still at the research level. Please either tone down (change) the title or provide additional references supporting the title.

Authors response:

After carefully reading this reviewer’s comment, we agree with his point, the original title was a bit misleading. The objective of this section as the reviewer realized is the presentation of novel approaches in non- or semi- controlled conditions. That was the original objective of this section. Because of this, we have retitled this section as "Experimental Robotic Applications Using Polarimetry" to better align with the content.

3) Related to item 2): while sections 2.4, 8-10 provide sobering comments about the current state of the polarimetric imaging, it could be helpful to include a general comment about the technology readiness level (TRL) in the introduction. From the manuscript, a reader may infer TRL = 5 or 6 at most (this covers hardware and relevant software).

Authors response:

We appreciate this remark, and we have added a paragraph on Technology Readiness Levels (TRL) at the end of the introduction (lines 101-106) to put the use of polarimetric imaging in context.

We believe that this addition aims to help the reader understand the maturity and applicability of this technology.

By detailing the TRL, we provide insights into the current stage of development, from basic research to full deployment, thereby clarifying the practical potential and limitations of polarimetric imaging in various applications.

Minor issues:

4) Please check if all referenced papers have a complete and correct quotation; some are incomplete (e.g., [25], line 799).

Authors response:

We reviewed and cleaned all the references to ensure accuracy and consistency.

5) Please check the language and rephrase where needed, e.g. line 668: “This lack of data cannot be overcome by standard data augmentation methods, which is why some work aims to augment polarimetric data.”

Authors response:

We have checked the language and some rephrases were addressed.

Specially, the sentence that pointed the reviewer is being rewrite as:

The absence of data cannot be solved by standard data augmentation methods.

This is the reason certain research focused on creating specific polarimetric data augmentation methods.

Also other revisions were made to the sentences in lines 26-35, 43-49, 52-53, and 66-67.

6) It is rather unusual to report in the review paper the unpublished results of our own ongoing research (not as a quotation of previous work but in the form of tables with actual results). The authors may consider whether extracting section 7 Experimentation and publishing it as a separate research paper (letter or full article) may be a better way to show their work.

Authors response:

We understand that it is a bit unusual to report experimental results in a review paper. But section 7 is intended to compare the chosen SOTA methods (rather our contribution) by using common criteria. In addition to that, reviewer 1 considers this kind of evaluation to guide readers to select the appropriate method.  Following the advice of reviewer 1, we included images (Figures 1 to 11) to enhance the comprehensibility of our experimental results. These figures provide a visual representation of the data.

These visual aids are intended to complement the text and tables explanations, offering a clearer and more intuitive understanding of the findings.

By presenting the results in textual, table and visual formats, we aim to cater to diverse learning preferences and improve the overall accessibility of our research.

*** Comments on the Quality of English Language ***

Please see item 5) in main Comments section.

Round 2

Reviewer 1 Report

Comments and Suggestions for Authors

accept